# Insight into Factors Influencing Wound Healing Using Phosphorylated Cellulose-Filled-Chitosan Nanocomposite Films

**DOI:** 10.3390/ijms222111386

**Published:** 2021-10-21

**Authors:** Marta Kędzierska, Sara Blilid, Katarzyna Miłowska, Joanna Kołodziejczyk-Czepas, Nadia Katir, Mohammed Lahcini, Abdelkrim El Kadib, Maria Bryszewska

**Affiliations:** 1Department of General Biophysics, Faculty of Biology and Environmental Protection, University of Lodz, 90-236 Lodz, Poland; katarzyna.milowska@biol.uni.lodz.pl (K.M.); maria.bryszewska@biol.uni.lodz.pl (M.B.); 2Euromed Research Center, Engineering Division, Euro-Med University of Fes (UEMF), Fès 30070, Morocco; blilidsara@gmail.com (S.B.); n.katir@ueuromed.org (N.K.); a.elkadib@ueuromed.org (A.E.K.); 3Laboratory of Organometallic and Macromolecular Chemistry-Composites Materials, Faculty of Sciences and Technologies, Cadi Ayyad University, Marrakech 40000, Morocco; m.lahcini@uca.ma; 4Department of General Biochemistry, Faculty of Biology and Environmental Protection, University of Lodz, 90-236 Lodz, Poland; joanna.kolodziejczyk@biol.uni.lodz.pl

**Keywords:** chitosan, cellulose, phosphorylated cellulose, bio-composites, wound-healing

## Abstract

Marine polysaccharides are believed to be promising wound-dressing nanomaterials because of their biocompatibility, antibacterial and hemostatic activity, and ability to easily shape into transparent films, hydrogels, and porous foams that can provide a moist micro-environment and adsorb exudates. Current efforts are firmly focused on the preparation of novel polysaccharide-derived nanomaterials functionalized with chemical objects to meet the mechanical and biological requirements of ideal wound healing systems. In this contribution, we investigated the characteristics of six different cellulose-filled chitosan transparent films as potential factors that could help to accelerate wound healing. Both microcrystalline and nano-sized cellulose, as well as native and phosphorylated cellulose, were used as fillers to simultaneously elucidate the roles of size and functionalization. The assessment of their influences on hemostatic properties indicated that the tested nanocomposites shorten clotting times by affecting both the extrinsic and intrinsic pathways of the blood coagulation system. We also showed that all biocomposites have antioxidant capacity. Moreover, the cytotoxicity and genotoxicity of the materials against two cell lines, human BJ fibroblasts and human KERTr keratinocytes, was investigated. The nature of the cellulose used as a filler was found to influence their cytotoxicity at a relatively low level. Potential mechanisms of cytotoxicity were also investigated; only one (phosphorylated microcellulose-filled chitosan films) of the compounds tested produced reactive oxygen species (ROS) to a small extent, and some films reduced the level of ROS, probably due to their antioxidant properties. The transmembrane mitochondrial potential was very slightly lowered. These biocompatible films showed no genotoxicity, and very importantly for wound healing, most of them significantly accelerated migration of both fibroblasts and keratinocytes.

## 1. Introduction

The skin, the largest organ of the human body, has many important functions. First, it creates a barrier against the penetration of pathogens as well as chemical and physical factors. It is also involved in regulating body temperature, perceives external stimuli, protects against water loss, affects the hormonal balance, and contributes to the immune system [1,2,3]. Due to the complexity and importance of these functions for the organism, the organ must function properly and, especially, must ensure its full continuity. Therefore, many mechanisms have evolved to ensure effective healing of wounds and skin defects. Proper wound healing protects organisms against pathogens that can cause local and systemic infections, including sepsis [4,5].

One of the problems in modern medicine is the process of healing chronic wounds. Each year, millions of people worldwide experience both acute and chronic skin injuries, and ~37 million people suffer from chronic wounds [6]. Chronic wounds most often affect the elderly. The rate of their occurrence is constantly increasing due to the aging of society and the ages of patients [7,8]. “Non-healing” wounds most often concern patients suffering from peripheral arteriosclerosis, diabetic foot, and people immobilized with bedsores [9].

Wound healing is a very complicated process due to the involvement of many factors and the complexity of the associated mechanisms. The repair process depends on many types of cells and factors, including fibroblasts, growth factors, cytokines and elements of the extracellular matrix [10]. During healing, complex reactions of chemically and biologically active substances take place as well as physical phenomena expressed by an increase in tensile strength and changes in skin elasticity.

Healing mechanisms can be divided into three main phases: debridement (hemostasis), cell proliferation (migration), and restoration (protein generation and wound contraction with scar formation) [11]. Damage to the skin is associated with the rupture of blood vessels and bleeding, which are inseparable features of a wound. Hemostasis and inflammation then occur.

Hemostasis, its main goal being to create a lamellar plug that protects the wound against external factors and stops bleeding, begins immediately after injury. At the same time, monocytes, neutrophils, and mast cells diffuse into the plug-forming site. The hemostatic response, including the activation of blood platelets, triggers the plasma coagulation cascade and generation of the thrombin enzyme, which are important elements that stimulate tissue remodeling and wound healing [12]. The clot is a temporary structure that allows the migration of biomolecules and is the basis of wound healing factors [5]. One of main cellular factors involved in the healing mechanism is platelet-derived growth factor (PDGF) in the blood. The essence of its presence is the activation of fibroblasts and macrophages, triggering a further stage of the tissue repair process by creating the extracellular matrix (ECM) [13,14].

The next stage is cell proliferation and reconstruction of connective tissue containing collagen fibers. Macrophages determine the transition from the purification phase to the multiplication phase. 

Proliferation leads to the formation of a base called granulation tissue, on which new epidermis is reconstructed. Within two weeks of injury, mast cells become active, stimulating the rapid multiplication of keratinocytes and fibroblasts. The substrate for this reconstruction of the epidermis consists of a network of collagen fibers, glycoprotein contained in the extracellular matrix (fibronectin), and hyaluronic acid. An integral process in the proliferation phase is angiogenesis [15,16]. In the final stage of wound healing, the extracellular matrix is restructured, and type I collagen is produced. The wound is completely closed with the participation of contractile myofibroblasts [2].

Current therapeutic strategies are insufficiently effective, making it necessary to focus on new therapeutic approaches and develop technologies for treating both persistent short- and long-term wounds. In this framework, the inherent properties of natural polysaccharides seem to be attractive; their use has already stimulated extensive research to find improved devices [17,18,19,20,21,22].

Chitosan extracted from crustaceans is one of the most interesting polysaccharides, owing to the abundance of its source, the presence of nitrogen in the skeletal structures, and is ease of shaping into hydrogels, porous beads, and transparent films [23,24]. Its potential drawback lies in its poor mechanical properties, which can be circumvented by adding a tiny amount of nanometric filler (e.g., graphene oxide, ceramic metal oxide, clay, calcium carbonate, metals, and hard carbon) [25,26,27,28]. We previously investigated the entrapment and growth of different objects inside chitosan films to access highly reactive bioplastics [29,30,31,32]. With the aim of avoiding any undesirable or side effects that might emanate from the filler nature, we further explored the use of naturally abundant and biodegradable micro- and nanocellulose as fillers, allowing access to fully compatible cellulose-filled chitosan nanocomposites [33]. Interestingly, phosphorylated cellulose fillers also enhance the antibacterial activity of chitosan films, making these materials active without the entrapment of silver nanoparticles, zinc oxide clusters, or any other costly synthetic metals. Additionally, these films are transparent, which is very convenient when used as wound dressings to directly observe the wound and accurately monitor its healing. Herein, we investigated the use of phosphorylated and native cellulose-filled chitosan biocomposites as potential materials to accelerate wound healing. These biomaterials were tested for their hemostatic properties, cytotoxicity against two cell lines (fibroblasts and keratinocytes), antioxidant properties, and influence on cell migration.

## 2. Results and Discussion

For our study, we selected seven chitosan-based films (Figure 1). One of them, denoted as CS, was built by casting a colloidal solution of chitosan; it served as a benchmark. We also added 3 wt% microcrystalline cellulose (MCC) and nanosized cellulose (CNC) to the starting chitosan cellulose to create CS@MCC and CS@CNC, respectively. Herein, any comparison should reflect the pivotal role played by the size of the filler and could highlight interest in downsizing cellulose to the nanoscale. We also modified both MCC and CNC with POCl_3_ to produce P-MCC and P-CNC and used cyclo- triphosphazene ring to access PN-MCC and PN-CNC [34]. These phosphorylated celluloses were also used as fillers to build novel cellulose-filled chitosan films. Any discrepancy in material performance could be attributed to filler functionalization. Detailed characterization of these materials has already been described in [33,34].

### 2.1. Hemostatic Properties

The first stage was done to evaluate the influence of cellulose biocomposites on the activation of the extrinsic and the intrinsic pathways of the blood plasma coagulation system. The effects of the biocomposites were determined based on well-known diagnostic biomarkers, i.e., blood clotting times. Prothrombin (PT), thrombin (TT), and the activated partial thromboplastin time (aPTT) were determined after 15 min of incubation with the tested films. Figure 2 shows that all of these coagulometric parameters were shortened compared with control blood plasma, which indicated that the composites accelerated blood coagulation.

Data on prothrombin times after incubation with cellulose films are provided in Figure 2A. The control plasma samples (i.e., blood plasma untreated with biocomposites) coagulated after 15.3 ± 1.4 s. All biocomposites reduced the time similarly to 14–13 s. Figure 2B shows thrombin times. For the control sample, the time was 17.7 ± 2.2 s. Incubation with biocomposites reduced this by ~4 s. The most efficient procoagulant activity in the TT measurements occurred in samples incubated with CS@P-MCC-f. Another time measured was the aPTT, which indicated the activity of the intrinsic coagulation pathway. The control took 54.6 ± 5.5 s, and the incubation of plasma with biocomposites significantly shortened the clot formation time (Figure 2C). The highest level of clotting activity occurred with the CS@PN-CNC-f composite, for which the aPTT time was 35.6 ± 4.5 s. Similar results were found by Li et al. [35] who tested the influence of collagen with oxidized microcrystalline cellulose on hemostatic properties. aPTT and TT of sponge collagens with 0.25% cellulose (M2) had the lowest levels among the materials tested, being significantly reduced compared to the normal saline group. The results show that a cellulose supplement can shorten the aPTT time and activate blood plasma clotting factors (VIII, IX and XI and XII), which corresponds to M2 promoting factor XII activation. The data also imply that M2 has a direct impact on the intrinsic coagulation pathway but not on the extrinsic pathway. A reduction in TT indicated that the conversion of plasma fibrinogen into a fibrin clot in the samples of the M2 group was significantly increased compared with the control group. In summary, the addition of cellulose to the tested material could be directly involved in stimulating the intrinsic coagulation pathway, thereby accelerating blood plasma clotting.

Cheng et al. [36] studied the hemostatic mechanism for oxidized microcrystalline cellulose and its composites. They showed that ORC (Oxidized Regenerated Cellulose), OMCC (Oxidized Microcrystalline Cellulose), and the hemostatic composite affected the activation of coagulation factors VIII, IX, XI, XII but did not affect factors III and VII, suggesting that only the intrinsic blood coagulation pathway was activated. This activity can significantly accelerate the activation of blood coagulation factor XII and promote the generation of thrombin [37]. In the presence of thrombin, soluble fibrinogen polymerizes and turns into a fibrin clot. The formation of a fibrin network on the surface of the damaged blood vessel wall is crucial for filling the injury, stopping bleeding, and modulating the activity of the coagulation and fibrinolytic proteins as well as stimulating wound healing [38]. As in the case of the tested cellulosic biomaterials, this investigation focused on a sponge based on large, mesoporous silica nanoparticles (MSN) and *N*-alkylated chitosan (AC) [39]. aPTT was significantly shortened by MSN, MSN-GACS (mesoporous silica nanoparticles with a glycerol-modified *N*-alkylated chitosan sponge), and kaolin, whereas AC had no influence on aPTT. In addition, the PT of all of these agents was unchanged, indicating that MSN-GACS does not significantly affect the extrinsic coagulation pathway. SiO_2_ and kaolin can induce activation of the intrinsic coagulation pathway by activating plasma coagulation factor XII. Therefore, the results suggest that the MSNs in MSN-GACS, rather than AC, freely contact the blood components and activate the intrinsic pathway. The coagulation potential of AC could depend on its positive charge, which can be adsorbed on the cytomembranes of blood cells and by some proteins [40].

### 2.2. Measurement of the Total Antioxidant Capacity

The antioxidant capacity of chitosan-cellulose biocomposites was analyzed on the basis of ABTS’s radical scavenging capacity, as shown in Figure 3 [41,42].

All nanocomposites tested were found to have antioxidant properties after 15 min of incubation. The nanocomposite with the highest ABTS radical scavenging capacity was the best antioxidant. In the case of the films, free radicals were most effectively removed by the chitosan film, where the scavenging capacity was 83.5%. Nanocomposites containing cellulose in their structure also show antioxidant activity; however, the radical scavenging values were lower compared with CS-f. Nanocomposites reinforced with phosphorylated micro- and nanocellulose (CS@P-MCC-f and CS@PN-MCC-f) were found to have similar antioxidant properties to the nanocomposite without modification, CS@MCC-f. The results are consistent with those presented in other reports, suggesting that cellulose has antioxidant properties [43]. Materials containing nanocellulose have greater antioxidant capacity than those containing microcrystalline cellulose. The highest value for CS@PN-CNC-f was 46%. These results suggest that adding cellulose to chitosan may partially reduce its antioxidant properties. Others have tested the antioxidant properties of different cellulose-containing composites. Zhang et al. [44] showed that cellulose nanocrystals (CNC) and cellulose nanofiber (CNF) are excellent controlled release agents and stabilizers that significantly increase the antioxidant and antibacterial properties of edible food-packaging films. Others have also shown antioxidant properties of composites containing cellulose; however, they were often increased by adding other substances, e.g., Rosemary and Aloe Vera essential oils, which contain polyphenols [45] or melanin nanoparticles [46].

### 2.3. Cell Viability

The cytotoxicity of cellulose-filled-chitosan biocomposites was assessed by cell viability studies using BJ and KERTr cell lines after 24 h of incubation with biocomposites. Cell viability was assessed with the MTT assay. The percentages of viable cells are given relative to control cells incubated without biomaterials (the control being taken as 100%) in Figure 4. All of our tested composites, except for CS-f, showed statistically significantly decreased viability for both cell lines. However, the viability of the cells was not <70%. The decrease in viability depended on the composition of composites and the type of cells. Keratinocytes were more sensitive than fibroblasts (Figure 4). The biggest decrease in KERTr viability was found with the three microcrystalline cellulose composites. For CS@MCC-f, CS@P-MCC-f and CS@PN-MCC-f, the viabilities ranged from 70 to 72%. The least toxic were the CS@CNC-f and CS-f composites, which reduced viability by ~15%. For fibroblasts, the most toxic film (CS@P-CNC-f) decreased viability to ~76%. In contrast, the viability of BJ treated with CS-f remained at the control level (Figure 4A).

There is information on the cytotoxicity of biomaterials containing nanocellulose. Bionanocomposites that do not decrease viability below 80% can be considered noncytotoxic. Poonguzhali et al. [47] assessed the effect of chitosan-PVP-nanocellulose composites on fibroblasts (NIH-3T3). All of the composites decreased in viability to 40–70% after 24 h of incubation and to 60–80% after 3 and 5 days. An increase in viability proved the good compatibility of the composites, because after 3 and 5 days, the cells began to proliferate. Naseri et al. [48] used porous nanocomposite electrospun mats based on chitosan-cellulose nanocrystals for wound healing. The in vitro cytocompatibility of cellulose nanocrystals and electrospun mats (MCNC_HCl_ and XMCNC_HCl_) was measured in a direct contact system using adipose-derived stem cells (ASC) and the L929 cell line. The biomaterials were stained with MTT. Cells in contact with all of their materials had similar morphologies to the negative (non-cytotoxic) control. There was no zone of cell growth inhibition; therefore, the biomaterials were considered noncytotoxic. Some previous studies also showed no cytotoxicity for nanocellulose and nanochitin [49,50,51].

In our study, the slight decrease in metabolic activity compared with the control was probably unrelated to cell death; the decrease could have been related to decreased cell proliferation. One possible explanation for the decreased proliferation may have been increased mechanical stress caused by polysaccharides, which could have affected the proliferative capacity of other cell types in 3D matrices [52]. A bigger reduction in metabolic activity by keratinocytes may indicate that these types of cells are more sensitive than fibroblasts, consistent with the findings of Burd et al. [53]. Similar results were shown by Čolić et al. [54]. CNF (cellulose nanofibrils) material was noncytotoxic to keratinocytes. The cells retained the same morphology as the control cells, indicating the suitability of CNF as a wound dressing material They also showed reduced metabolic activity and cell proliferation but low cytotoxicity from CNF. Cellulose materials have previously been shown to reduce cell proliferation without affecting the viability of keratinocytes [55]. CNF (50 µg/mL) did not affect fibroblasts or keratinocytes over a 24 h incubation period, showing that the potential leakage of CNFs in a wound could be acceptable [56].

### 2.4. Generation of Reactive Oxygen Species

To check whether reactive oxygen species (ROS) reduce the viability or proliferative capacities of BJ and KERTr cells, their level was determined after 24 h of incubation with our tested materials, the results being related to the control (100%). Only one of the tested films (SC@P-MCC-f) significantly increased the level of ROS in BJ cells. The other tested films did not significantly affect ROS in cells or might have reduce their concentration in some cases (Figure 5). Regarding fibroblasts (Figure 5A), the most significant decrease in ROS occurred in samples incubated with CS@P-CNC-f and CS@CNC-f.

Similar results were obtained for modified cellulose nanofibrillation by Aimonen et al. [57], who investigated whether wood-derived nanofibrillated cellulose (NFC) induces intracellular ROS production. Only U-NFC (unmodified) induced a significant increase in ROS formation compared to the negative control at 500 µg/mL after 24 h of exposure, whereas C-NFC (carboxymethylated), H-NFC (hydroxypropyltrimethylammonium), P-NFC (phosphorylated), and S-NFC (sulphoethylated) did not significantly increase the level of ROS for any doses or exposure times.

### 2.5. Assessment of Mitochondrial Membrane Potential (ΔΨm)

Alterations to the mitochondrial membrane potential (ΔΨm) after incubation with chitosan-cellulose biocomposites were monitored using a JC-1 fluorescent probe technique. Experiments on our two cell lines, BJ and KERTr, showed that unmodified chitosan film (CS-f) slightly, but significantly, increased ΔΨm, which suggests that the film does not affect proapoptotic activity (Figure 6). On the other hand, chitosan-cellulose films did not change ΔΨm, or some of them just gave slightly lower potentials. For fibroblasts incubated with CS@P-MCC-f, the lowest value was 90% of the control value, and for keratinocytes, the biggest decrease was with the CS@CNC-f composite, ΔΨm (88%).

As suggested by others, biomaterials containing modified forms of cellulose may lower the ΔΨm by generating mitochondrial ROS [58,59,60]. Sunasee et al. [61] showed that the cationic nanocrystalline cellulose (CNCs) derivative induces NLRP3 inflammasome-dependent IL-1β secretion associated with mitochondrial ROS production. CNC-AEMA2 (aminoethylmethacrylamide) was associated with the biggest loss in ΔΨm, indicating a decrease in the red/green ratio compared with untreated cells. Depolarization of the mitochondrial membrane can directly impact ATP production by mitochondria. Therefore, intracellular and extracellular levels of ATP were also measured. Both CNC-AEMA1 and CNC-AEMA2 decreased intracellular ATP in J774A1 (mouse macrophage) cells; however, CNC-AEMA2 gave a more pronounced negative effect than CNC-AEMA1, as reflected by a significant increase in the extracellular ATP content.

We found a relationship between the ΔΨm and ROS results for CS@P-MCC-f. This film gave the highest increase in ROS inside BJ cells and led to the largest decrease in ΔΨm, which explains the mechanism of cell membrane depolarization, as the production of intracellular reactive oxygen species interferes with cellular ATP levels, thereby reducing the mitochondrial potential [62,63].

### 2.6. Migration of Fibroblasts and Keratinocytes

Another important aspect is the influence of chitosan-cellulose composites on the migration process of BJ and KERTr cells. The ability of keratinocytes and fibroblasts to migrate to the wound environment from adjacent areas also determines the success of the healing process. The level of cell migration was determined by the migration assay using ThinCertTM (Figure 7). Among the biomaterials analyzed, CS@CNC-f and CS@MCC-f most strongly increased fibroblast migration after 24 h of incubation. These composites increased cell migration by one and a half times compared with untreated cells. Composite CS@PN-CNC-f had the smallest influence on the migration of fibroblasts. In the case of keratinocytes, migration was lower than with fibroblasts. The highest level of keratinocytes migrated in samples treated with CS@MCC-f was similar to fibroblasts, but the value was lower, ~126%. Interestingly, one of the CS@PN-CNC-f films delayed the migration of keratinocytes compared with the control. It can be concluded that both the microcrystalline and nanocrystalline forms of cellulose have positive effects on migration; however, modification with P- (phosphorylated) and PN- (cyclotriphosphazene) groups reduce the migration effect. Wang et. al. [64] investigated whether bacterial cellulose (BC)/gelatin membranes with electric field stimulation affect cell migration, thereby accelerating wound healing. In particular, a 40% stretched BC/gelatin membrane promoted the adhesion, orientation, and migration of NIH3T3 cells. The aligned BC/gelatin membrane synergistically directed the migration of NIH3T3 cells and significantly improved wound healing by accelerating wound closure, increasing the granulation thickness, collagen deposition, and angiogenesis. These findings suggest that a combination of 40% stretched BC/gelatin with electric field stimulation may be a promising therapeutic strategy to guide cell migration for improving wound healing.

Research by Bacakova et al. [65] aimed to improve a clinically used carboxymethylcellulose (Hcel^®^ NaT) wound dressing by coating it with fibrin and pre-seeding it with skin fibroblasts to create a cell carrier with the potential to deliver skin cells to a wound. This novel cell-enriched dressing is expected to improve the healing capacity of deep wounds. After degradation of the fibrin coating by cells, their cellulose scaffolds would become less attractive than the wound bed for cell adhesion, and thus, spontaneous release and migration of cells from these scaffolds could be expected. The scaffolding could then be easily removed, and thus scaffolds based on fibrin-modified cellulose can serve as cell carriers for skin wounds. A similar phenomenon was seen with human keratinocytes grown on poly-(2-hydroxyethyl methacrylate) plates used clinically to treat severe burns [66].

### 2.7. Genotoxicity

For testing materials that may have potential use in medicine, it is important to check their activities from different aspects, notably their genotoxicity. For this purpose, the comet test was used, which gives an answer to whether composites induce single- or double-stranded DNA breaks. Genotoxicity of materials may result from direct interactions with DNA or from an indirect response induced by several factors, including surface stress, through the direct influence of particles on DNA, the release of toxic ions from soluble nanoparticles, or the generation of oxidative stress [67,68,69]. The percentage of DNA in the tail is used to describe damage in the test specimens. The results shown in Figure 8A,B concern the DNA content in the tails of fibroblasts and keratinocytes, respectively, after 24 h of incubation with chitosan-cellulose biocomposites. Negative controls were untreated cells, and positive controls were cells treated with H_2_O_2_. All films significantly increased the tail moment. The greatest tail moment was found for cells treated with CS@P-MCC-f. CS@P-MCC-f also slightly increased the level of ROS, which might have caused oxidative stress and led to DNA damage. However, these changes did not exceed 10%, so it can be concluded that the tested films are non-genotoxic. Genotoxicity measurements of cellulose-containing biocomposites were carried out by Coelho et al. [70], who checked the toxicity of bacterial cellulose membranes functionalized with hydroxyapatite and the antibone morphogenetic protein 2 (BC-HA) on a murine osteoblast line, MC3T3. The percentage of DNA in the tail determined using the Comet Assay revealed that BC-HA is non-genotoxic compared with its negative control (NC); the percentage of DNA in the tail was found to be approximately 3%. They also tested the toxicity of therapeutic contact lenses based on bacterial cellulose with coatings to provide transparency. The assessment showed genotoxicity in only one case, but this was due to diclofenac sodium. None of the lenses tested had a mutagenic effect [71]. Moreira et al. [72] also confirmed that cellulose is nongenotoxic; they concluded from the comet test that cellulose nanofibers (NFs) do not induce DNA strand breaks or crosslinks.

## 3. Materials and Methods

### 3.1. Materials

Cellulose-filled chitosan nanostructured films were prepared in accordance with previously described procedures [33]. The commercially available reagents and solvents phosphoryl chloride, hexachlorotricyclophosphazene, ethanol (99.8%), tetrahydrofuran (97%), and acetic acid (98%) were purchased from Across and Sigma-Aldrich (St. Louis, MO, USA). Chitosan (190–310 kDa) and 85% deacetylation degree were purchased from Sigma-Aldrich (Hamburg, Germany). Phosphate-buffered saline (PBS) was purchased from BioShop (Burlington, ON, Canada). Glutaraldehyde 25% and osmium tetroxide 4% solution were purchased from Agar Scientific (Stansted, UK). Absolute ethanol was purchased from EMSURE (Darmstadt, Germany). Microcrystalline cellulose (MCC, CAS 9004-34-6) synthesized from cotton linters was purchased from Sigma-Aldrich. Cotton wool used for the synthesis of nanocellulose was purchased from Fisher Scientific (Hampton, NH, USA). Ultrasonication involved a VWR ultrasonic cleaner (USC-THD: Power 9 VWR International GmbH, Vienna, Austria).

The human fibroblast BJ (CRL-2522) cell line and human keratinocyte CCD 1102 KERTr (CRL-2310) were purchased from American Type Culture Collection ATCC^®^ (Manassas, VA, USA). Keratinocyte serum-free medium with added keratinocyte supplements, including bovine pituitary extract (BPE), human recombinant epidermal growth factor (EGF), fetal bovine serum (FBS), and Dulbecco’s modified Eagle’s medium (DMEM), was purchased from Gibco, Thermo Fisher Scientific (Waltham, MA, USA). Blood from healthy donors was obtained from the Regional Blood Donation and Blood Treatment Center in Lodz, Poland. Dimethyl sulfoxide (DMSO) 3-(4,5-2-yl)-2-5-diphenyl tetrazolium bromide (MTT), 5′,6,6′-tetrachloro-1,1′,3,3′-tetraethyl-imidacarbocyanine iodide (JC-1), 2,7-dichlorodihydrofluorescin diacetate (H_2_DCFDA), 2,2′-azinobis(3-ethylbenzothiazoline-6-sulfonic acid) diammonium salt (ABTS), potassium persulfate (di-potassium peroxdisulfate), phosphate buffered saline (PBS) tablets, fetal bovine serum, and trypsin were purchased from Sigma-Aldrich (Saint Louis, MO, USA). Commercially available reagents for the determination of clotting times (Dia-PT and DiaPTT) were purchased from Diagon (Budapest, Hungary). The thrombin enzyme was provided by Biomed (Lublin, Poland).

Fresh human blood plasma for hemostatic assays derived from buffy coats (from healthy volunteers) was purchased from the Regional Centre of Blood Donation and Blood Treatment in Lodz (Poland).

The genotoxicity study used a fluorescent dye that strongly binds to DNA by DAPI intercalation (Gibco, Thermo Fisher Scientific, Waltham, MA, USA). Membrane culture inserts for 24-well plates, PET, and 8 µM pores to check cell migration were purchased from Biokom (Janki, Poland).

All other chemicals used were of analytical grade, and solutions were prepared using water purified by the Mili-Q system.

### 3.2. Measurements of Prothrombin Time (PT)

Activation of the extrinsic and intrinsic pathways of blood coagulation [73] was determined using the Optic Coagulation Analyzer K-3002 (KSELMED, Grudziadz, Poland) [74,75]. In measurements of the PT, human plasma (1.5 mL) was incubated with cellulose films in the form of squares (0.5 × 0.5 cm) for 15 min at 37 °C and then sampled in a 50 µL coagulometric cuvette with 50 µL thromboplastin (i.e., Dia-PT reagent; DIAGON, Budapest, Hungary; a commercial preparation was dissolved in 2 mL deionized water) and incubated for 1 min at 37 °C on a block heater. After incubation, the cuvette was transferred to the analyzer, and 50 µL of 25 mM CaCl_2_ was immediately added.

### 3.3. Measurements of the Thrombin Time (TT)

Human plasma (1.5 mL) was incubated with cellulose biomaterials in the form of squares (0.5 × 0.5 cm) for 15 min, sampled into a 50 µL measuring cuvette, and incubated for 1 min at 37 °C on a block heater. The cuvette was transferred to the measuring holes and 100 µL thrombin (Biomed, Lublin, Poland) was added (final concentration 1 U/mL). The thrombin time (TT) was determined coagulometrically (Optic Coagulation Analyser K-3002; KSELMED, Grudziadz, Poland).

### 3.4. Activated Partial Thromboplastin Time (aPTT)

Activation of the intrinsic blood coagulation pathway was determined using the Dia-PTT reagent (i.e., cephalin preparation for activated partial thromboplastin time measurements dissolved in 4 mL deionized water and incubated at 37 °C for 30 min). A cuvette was placed in the coagulometer thermostat (Optic Coagulation Analyzer K-3002; KSELMED, Grudziadz, Poland) and 50 µL of plasma previously incubated for 15 min with cellulose biocomposites was introduced to it. The dia-PTT reagent was added to the plasma in the cuvette. The mix was incubated at 37 °C for 3 min. Then, 50 µL of 0.025 M CaCl_2_ was added to the cuvette prior to measurement. Following the addition of CaCl_2_, further clotting factors were activated, leading to the formation of a blood plasma fibrin clot.

### 3.5. Measurement of the Total Antioxidant Capacity

A colored blue-green solution of the radical ABTS (ABTS^•+^) is formed by the reaction between ABTS and K_2_O_8_S_2_. ABTS and potassium persulfate were dissolved in distilled water to final concentrations of 7 and 2.45 mM, respectively. These two solutions were mixed and left to stand in a dark at room temperature for 16 h before use in order to produce the ABTS radical (ABTS^•+^). The absorption spectrum of ABTS^•+^ is characterized by the presence of several maxima for the wavelengths 415, 645, 734, and 815 nm. Adding an antioxidant to the solution reduces the concentration of the ABTS radical, discoloring the solution [76]. The ABTS radical solution was diluted with distilled water to an absorbance of 0.7–0.9 at 734 nm. Biocomposites were added to the diluted ABTS^•+^ solution (1.5 mL) and incubated for 15 min, and the absorbance was read using a spectrophotometer. The percentage of reaction inhibition was calculated from the formula (1):ABTS radical scavenging (%) = (1−A/A_0_) × 100%(1)
where:

A_0_—represents the absorbance of the control (only ABTS) at 734 nm

A—represents the absorbance of the sample (ABTS + film) at 734 nm

### 3.6. Cell Culture

Cells were cultured using regular practices. Adherent BJ cells were grown in DMEM growth medium, whereas the KERTr cells were grown in Keratinocyte-Serum Free medium (Gibco). DMEM was supplemented by increased concentrations of vitamins and amino acids, as well as pyruvate and glucose. DMEM was supplemented with 10% inactivated FBS and an antibiotic (1% penicillin). Keratinocyte-Serum Free (Gibco) medium was supplemented with Keratinocyte Supplements, including Bovine Pituitary Extract (BPE, Gibco) and human recombinant epidermal growth factor (EGF, Gibco). Both cultures were carried out under standard conditions in a CO_2_ incubator (37 °C, 95% air and 5% CO_2_, 100% relative humidity). Cells of both lines were maintained in the logarithmic growth phase by regular passage into new culture flasks after the cells had reached ~80% confluence. Monolayers were washed sequentially with an isotonic saline solution (0.9%), and the cell monolayers were trypsinized by adding 0.25% trypsin with EDTA and KERTr by adding a 0.25% TrypLE solution (Gibco) in the appropriate volumes. Cells were incubated for 3–5 min in a CO_2_ incubator and checked under a microscope. After detachment, DMEM culture medium was added to BJ and Keratinocyte-Serum Free was added to KERTr, respectively. Cell viability was checked at each passage using Trypan blue, which enters cells with damaged cell membranes, making them blue. A small amount of the cell suspension (~10 µL) was mixed 1:1 with Trypan blue and applied to the plates. During measurement, the total cell density and the amount of dead and living cells per 1 mL of suspension were determined.

### 3.7. Cell Viability Assay (MTT Assay)

The cytotoxicity of chitosan-cellulose films was measured by the MTT assay, which is based on cellular reduction of the soluble yellow dye 3-[4,5-tetrazolium salt) dimethylthiazol-2-yl]-2,5-diphenyltetrazolium bromide (MTT) by mitochondrial dehydrogenases to water-insoluble purple formazan in living cells. Therefore, the amount of formazan crystals is proportional to the number of living cells, because the dehydrogenases are inactive in dead cells [77,78].

The BJ cells were seeded in flat-bottomed 24-well plates at concentrations of 5 × 10^4^ (BJ) and 10^5^ (KERTr) cells in 400 µL medium per well. After 24 h, they were treated with films in the form of 0.5 × 0.5 cm squares, with the controls being untreated. The plates with biocomposites were incubated for 24 h under culture conditions before the MTT test was used. The medium was removed from the wells, and the cells were washed twice with PBS (200 µL). MTT reagent (200 µL) was added to the wells and incubated for 3 h at 37 °C. MTT was aspirated, and 400 µL DMSO was added to each well. The absorbance at 570 nm was measured spectrophotometrically (BioTek, Synergy HTX multi-mode reader, Winooski, VT, USA). MTT tests were repeated in 6 separate experiments. The percentage of viability was calculated using the formula (2):% Viablility= A_s_/A_c_ × 100%(2)
where: A_s_ is the absorbance of the sample and A_c_ is the absorbance of the samples control (untreated cells).

### 3.8. Generation of Reactive Oxygen Species 

H_2_DCFDA is a form of reduced fluorescein that freely penetrates cells, where it is hydrolyzed to its non-fluorescent form, H_2_DCF, by intracellular esterases. Subsequently, H_2_DCF is oxidized by ROS to 2′,7′-dichlorofluorescein (DCF). DCF is localized in the cytoplasm and is strongly fluorescence, its intensity being proportional to the concentration of ROS [79,80,81].

BJ cells were seeded in black 96-well plates at 1.25 × 10^4^ per well and KERTr cells at 2.5 × 10^4^ per well. After 24 h of incubation with the biocomposites, the medium was removed, and the cells were washed with PBS before 50 µL 2 μM H_2_DCFDA was added. The probe plate incubation time was 15 min. The solution was recovered, and 50 µL of PBS was added per well. Samples were analyzed using a Fluoroscan Ascent FL microplate reader (BioTek, Synergy HTX multi-mode reader) with an excitation wavelength of λ_ex_ = 495 and emission wavelength of λ_em_ = 529.

### 3.9. Assessment of the Mitochondrial Membrane Potential (ΔΨm)

One of the parameters that proves mitochondrial dysfunction is a decrease in ΔΨm. The JC-1 probe (5,5-iodide,6,6′-tetrachloro-1,1′,3,3′-tetraethylbenzimidazolycarbocyanine) was used to determine ΔΨm. The probe is a positively charged lipophilic fluorescent marker. Normal cells with appropriate potentials ensure more efficient functioning of metabolic pathways, providing the cell with more ATP, so ΔΨm values range from −120 to −180 mV. In damaged cells, energy production in the form of ATP is decreased, and membrane depolarization—a decrease in the potential—occurs. The primary location for the probe is the mitochondrial matrix. JC-1 comes in monomer and dimer forms. The dimeric form occurs when the mitochondrial membrane is polarized, with a high ΔΨm. Red fluorescence is then emitted. In the case of the monomeric form, the value of ∆Ψm is lower, and the characteristic fluorescence is green [82].

BJ cells were plated in black 96-well plates at 12.5 × 10^3^/well and KERTr cells were plated at 25 × 10^3^. After incubation for 24 h with biocomposites, the medium was removed. After washing with PBS, 50 µL of JC-1 (1 µM) was added. The plate was incubated for 30 min in the dark, and the solution was removed from the wells. PBS (50 µL) was added to each well and measured. In the analysis, specialized filters were used to measure the fluorescence of monomers (λex = 485 nm, λem = 538 nm) and dimers (λex = 530 nm, λem = 590 nm). From the measurement on a Fluoroscan Ascent FL microplate leader (BioTek, Synergy HTX multi-mode reader), the fluorescence coefficient was calculated (3):Ψm = Fd/Fm(3)
where Ψm is the transmembrane mitochondrial potential directly proportional to the fluorescence coefficient, Fd is the fluorescence of dimers, and Fm is the fluorescence of monomers.

### 3.10. Cell Migration

BJ and KERTr cells were starved overnight in serum-free medium with 0.2% bovine serum albumin (BSA) [83,84]. Harvested BJ cells were washed twice in PBS and resuspended in serum-free DMEM medium with 0.2% BSA to obtain an appropriate final concentration of 2.5 × 10^5^ cells/mL, whereas KERTr cells were resuspended in growth-factor-free medium to the same cell density. Twenty-four-well ThinCert™ cell culture inserts with 8 μM pores and translucent PET membranes were placed in the wells of a CELLSTAR^®^ cell culture plate. Six hundred microliters of DMEM medium with 10% FBS was added to each well of the BJ cell culture plate (lower compartment). For keratinocytes, Keratinocyte-Serum Free medium with growth factors was added to the lower wells. Two hundred microliters of BJ cell suspension with 450 μL of free-serum medium DMEM was added to each cell culture insert, and the same volumes of KERTr and free-growth factor medium Keratinocyte-Serum Free were used. The plate with inserts was incubated for 24 h in an incubator at 37 °C and 5% CO_2_ in air. The culture medium was removed from the ThinCert™ cell culture inserts, and the inserts were transferred to the wells of a freshly prepared 24-well plate containing 500 μL of Trypsin-EDTA per well. This plate was incubated for 10 min in a cell culture incubator at 37 °C and 5% CO_2_ with sporadic agitation. The inserts were discarded, and 200 μL of Trypsin-EDTA solution (now containing the detached migratory cells) was transferred from each well into a new well of a flat-bottom 24-well plate. Finally, the viability of the migrating cells was determined using the MTT test and measured spectrophotometrically at 570 nm.

### 3.11. Comet Assay

The alkaline version of the comet assay was carried out according to the procedure of Singh et al. [85] with slight modifications described by Blasiak and Kowalik [86]. Briefly, 100 µL (50,000 cells) of cell suspension was mixed with 10 µL of 0.75% low melting-point agarose (LMP) at 37 °C and spread on a normal agarose (NMP) pre-coated microscope slide. The slides with cells were covered with a coverslip and subsequently placed on an ice-cold surface to solidify for about 10 min. The coverslips were removed, and the slides were placed in cold lysing solution (2.5 M NaCl, 0.1 M EDTA, 10 mM Tris, 1% Triton X-100, 10% DMSO, pH 10; the last two components were added freshly). Lysis took 1 h at 4 °C in the dark. The slides were incubated in an electrophoretic buffer (300 mM NaOH, 1 mM EDTA, pH > 13) for 20 min to allow the unwinding of DNA before electrophoresis. Electrophoresis was run in the same buffer at 0.73 V/cm (28 mA) for 20 min to allow damaged DNA or fragments to migrate towards the anode. The slides were washed in water, drained, stained with 2 µg/mL DAPI, and covered with coverslips. Microscopic analysis was started after a minimum of 30 min. To prevent additional DNA damage, the whole procedure was conducted under limited light or in the dark. The comets were analyzed by an Eclipse fluorescence microscope (Nikon, Tokyo, Japan) attached to a COHU 4910 video camera (Cohu, Inc., San Diego, CA, USA) equipped with a UV-1 A filter block and connected to a personal computer-based image analysis system Lucia-Comet v. 4.51 (Laboratory Imaging, Praha, Czech Republic). The tail moment, as a measure of DNA damage in the graphic presentation, represents the mean of 50 images (comets) randomly selected from each sample of 3 individual experiments.

### 3.12. Statistical Analysis

Data are presented as the mean ± SD from a minimum of 3 sets of measurements. Statistical differences between the control and treatment groups were analyzed by one-way ANOVA followed by Tukey’s analysis. *p* < 0.05 was taken as statistically significant.

## 4. Conclusions

While the use of marine polysaccharides for wound healing has been intensely investigated, few reports have focused on deciphering the role of the filler used to reinforce the polysaccharide matrix in accelerating or delaying wound healing. We have consequently designed several nanocomposite films built from chitosan marine waste. The common threat with these materials is the use of cellulose as a filler. The use of icrometric versus nanometric cellulose and phosphorylated versus non-modified cellulose enabled the preparation of seven films. Our investigation allowed a comparative performance relationship based on the size of the filler and the functionalization type. Measurements of the hemostatic, cytotoxic, and genotoxic properties of cellulose-filled-chitosan films showed that these nanocomposites shorten the clotting times by affecting both the extrinsic and intrinsic coagulation systems and have antioxidant properties. They only slightly reduced the viability of human skin cells and were not genotoxic. In addition, most of the tested nanocomposites significantly accelerated the migration of both fibroblasts and keratinocytes, which is very important since increased skin cell migration promotes wound healing and accelerates scarring. Associated with the previously demonstrated antibacterial properties of phosphorylated cellulose-filled chitosan films, the results shown herein suggest that these transparent, fully degradable, and biocompatible nanocomposite films have the potential to be implemented in wound dressing devices. 

## Figures and Tables

**Figure 1 ijms-22-11386-f001:**
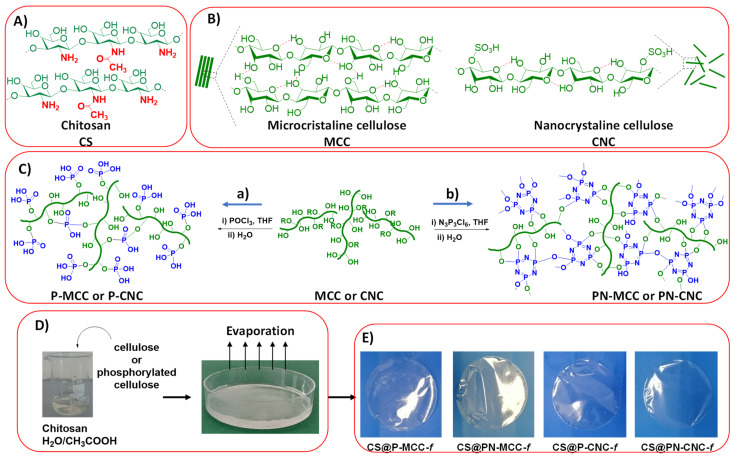
Multistep preparation of cellulose-filled chitosan films. (**A**) the chemical structure of chitosan used to generate flexible and transparent films; (**B**) the structure of the two cellulose fillers used to reinforce chitosan, namely microcrystalline cellulose (MCC) with tactoids and isolated tiny crystals of nanocellulose (CNC); (**C**) the use of two different phosphorylated reagents (POCl_3_ or N_3_P_3_Cl_6_) affords two kinds of phosphorylated cellulose (P-MCC/P-CNC and PN-MCC/PN-CNC); (**D**) mixing of chitosan and cellulose derivative followed by the introduction of the resulting solution in a petri dish and further evaporation of the solvent; (**E**) digital photos of the resulting cellulose-filled chitosan films, illustrating their transparency.

**Figure 2 ijms-22-11386-f002:**
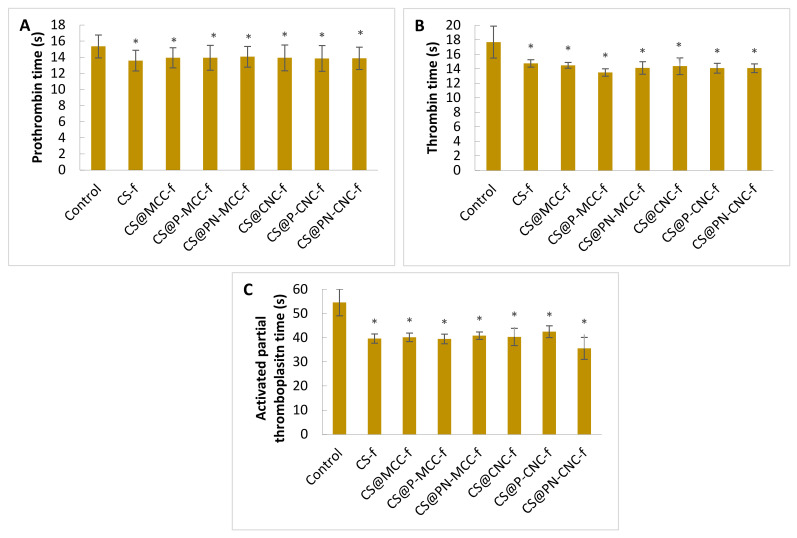
PT (**A**), TT (**B**), and aPTT (**C**) blood clotting times of control human blood plasma incubated with chitosan-cellulose biocomposites. *n* = 10, * *p* < 0.05.

**Figure 3 ijms-22-11386-f003:**
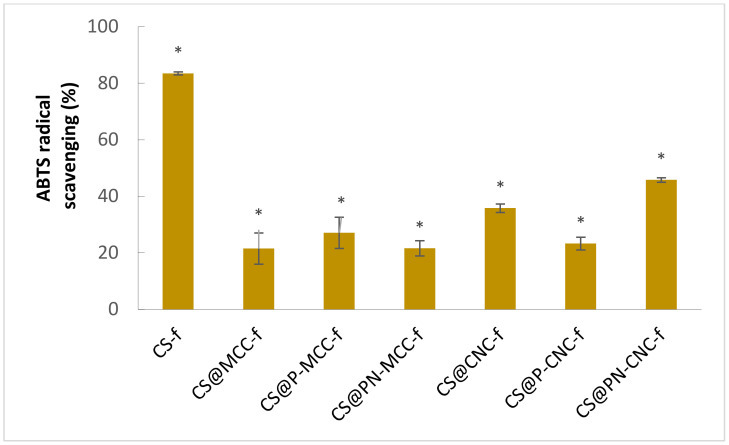
Antioxidant properties of chitosan-cellulose biomaterials after 15 min of incubation, expressed as a percentage of inhibition, *n* = 6, * *p* < 0.05.

**Figure 4 ijms-22-11386-f004:**
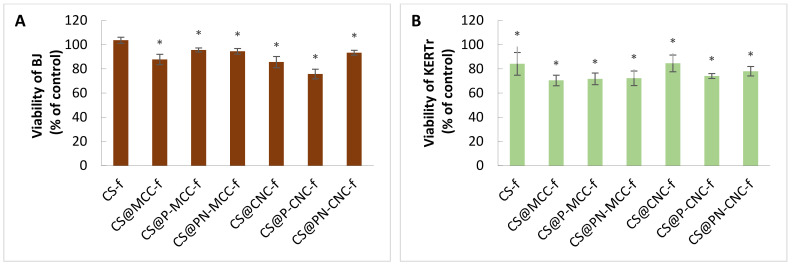
Viability of BJ (**A**) and KERTr (**B**) treated with chitosan-cellulose biocomposites for a 24 h incubation period, *n* = 6. * *p* < 0.05.

**Figure 5 ijms-22-11386-f005:**
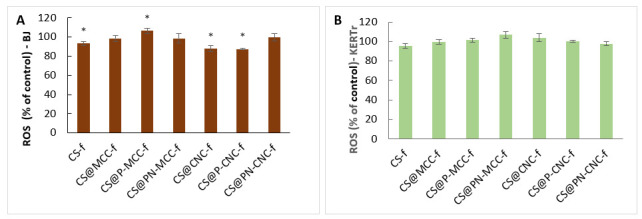
The content of reactive oxygen species in cells of BJ (**A**) and KERTr (**B**) incubated for 24 h with chitosan-cellulose biocomposites, *n* = 6. * *p* < 0.05.

**Figure 6 ijms-22-11386-f006:**
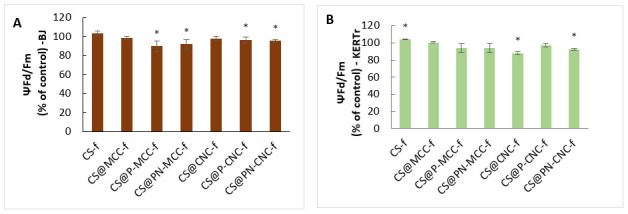
The transmembrane mitochondrial potential (ΔΨm) of BJ (**A**) and KERTr (**B**) incubated for 24 h with chitosan-cellulose biocomposites, *n* = 6. * *p* < 0.05.

**Figure 7 ijms-22-11386-f007:**
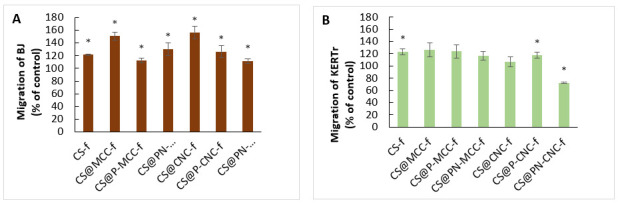
The migration of BJ (**A**) and KERTr (**B**) cells incubated for 24 h with chitosan-cellulose biocomposites, *n* = 6, * *p* < 0.05.

**Figure 8 ijms-22-11386-f008:**
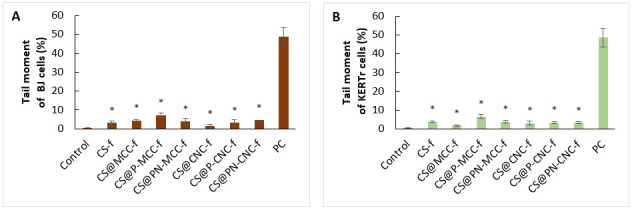
The genotoxicity of BJ (**A**), KERTr (**B**) cells incubated for 24 h with chitosan-cellulose biocomposites, *n* = 6, * *p* < 0.05.

## Data Availability

Not applicable.

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
