# Peer review of "Insight into Factors Influencing Wound Healing Using Phosphorylated Cellulose-Filled-Chitosan Nanocomposite Films"

_ijms, 2021, doi:10.3390/ijms222111386_

Round 1

Reviewer 1 Report

The authors Marta Kędzierska et al. have reported the article entitled “Insight on Factors Influencing Wound Healing Using Phosphorylated Cellulose-Filled-Chitosan Nanocomposite Films”. The manuscript was well written and explained thoroughly. The obtained results are significant enough to claim their statements. Based on the critical evaluation, the manuscript would be recommended for publication in the Int. J. Mol. Sci. Though, the manuscript needs to be improved on the following point. Thus, the manuscript needs a minor revision before publication.

  1. The manuscript needs a language correction.
  2. Figure 1 must be explained more elaborately; the current form is not easy to understand to the readers.
  3. The authors converted almost all of their data into chart form. For any of the biological studies, some of the images from the study would improve the quality of the manuscript.
  4. The authors are strongly advised to give images of the results for the following studies, 2.3. Cell viability, 2.4. Generation of reactive oxygen species, 2.5. Assessment of Mitochondrial Membrane Potential (ΔΨm), and wherever possible, provide the respective images.
  5. In all the bar diagrams, the error (red lines, y-axis) indications must be removed.
  6. Some of the important references related to bio toxicity must be included in the reference sections. ChemistryOpen 8 (5), 589-600; ChemMedChem 14 (5), 512-512; Molecules 24 (7), 1437.

Overall, the manuscript is sound and worth publishing with a minor revision

Author Response

University of Lodz

Faculty of Biology and Environmental

Protection, Pomorska st. 141/143, 90-236 Lodz

Lodz, 15.10.2021

RESPONSE TO REFEREES

To the manuscript, Submission ID ijms-1422171

Title: Insight on factors influencing wound healing using phosphorylated cellulose-filled-chitosan nanocomposite films

Dear Editor,

The authors are grateful for the Editor's comments as well as Reviewer advices. The manuscript has been improved accordingly to the reviewers remarks. We tried to address all the comments in this answer. The changes into the manuscript are highlighted in yellow color.

Response to Reviewer#1

Recommendations of the Reviewer have been taken into consideration as follows:

1.The manuscript needs a language correction

author response:  The article was subject to linguistic proofreading prior to the manuscript being made available to reviewers.

  1. Figure 1 must be explained more elaborately; the current form is not easy to understand to the readers.

author response: Figure 1 has been improved, I hope it is clearer and more understandable. Description has been changed: ,,Multistep preparation of cellulose-filled chitosan films. i) the chemical structure of chitosan used to generate flexible and transparent films. ii) the structure of the two cellulose fillers used to reinforce chitosan, namely microcrystalline cellulose (MCC) with tactoids and isolated tiny crystals of nanocellulose (CNC). iii) the use of two different phpsphorylated reagents (POCl3 or N3P3Cl6) affords two kinds of phosphorylated cellulose (P-MCC / P-CNC and PN-MCC/PN-CNC). iv) mixing chitosan and cellulose derivative followed by the introduction of the resulting solution in a petridish and further evaporation of the solvent. v) digital photos of the resulting cellulose-filled-chitosan films illustrating their transparency”

  1. The authors converted almost all of their data into chart form. For any of the biological studies, some of the images from the study would improve the quality of the manuscript

  1. The authors are strongly advised to give images of the results for the following studies, 2.3. Cell viability, 2.4. Generation of reactive oxygen species, 2.5. Assessment of Mitochondrial Membrane Potential (ΔΨm), and wherever possible, provide the respective images.

author response – points 3 and 4: Our research (2.3. Cell viability, 2.4.Generation of reactive oxygen species, and 2.5. Assessment of Mitochondrial Membrane Potential (ΔΨm), was based on reading the measurements on a BioTek microplate reader, Synergy HTX multi-mode reader. The results obtained were then converted into% control (control was 100%). We are not able to present the above results in the form of photos

  1. In all the bar diagrams, the error (red lines, y-axis) indications must be removed.

author response: The charts have been enlarged and refined to improve their quality and readability. In the loaded file, no red lines are visible on any graph.

  1. Some of the important references related to bio toxicity must be included in the reference sections. ChemistryOpen 8 (5), 589-600; ChemMedChem 14 (5), 512-512; Molecules 24 (7), 1437.

author response: Suggested publications have been cited in the paper as references. Markers in the text and references are marked in yellow.

Reviewer 2 Report

The paper Insight on factors influencing wound healing using phosphorylated cellulose-filled-chitosan nanocomposite films prepared by Marta Kędzierska et al., present novel and interesting results that deserve to be published after several improvements:

  1. Figure 1 - please increase the font size of the text. It is very hard to focus on it.
  2.  All charts must be aligned - one per row. Also, please use GraphPad or originlab software to create your charts.

After this minor revision the paper deserve to be published.

Author Response

University of Lodz

Faculty of Biology and Environmental

Protection, Pomorska st. 141/143, 90-236 Lodz

Lodz, 15.10.2021

RESPONSE TO REFEREES

To the manuscript, Submission ID ijms-1422171

Title: Insight on factors influencing wound healing using phosphorylated cellulose-filled-chitosan nanocomposite films

Dear Editor,

The authors are grateful for the Editor's comments as well as Reviewer advices. The manuscript has been improved accordingly to the reviewers remarks. We tried to address all the comments in this answer. The changes into the manuscript are highlighted in yellow color.

Response to Reviewer#2

  1. Figure 1 - please increase the font size of the text. It is very hard to focus on it.

author response: Figure 1 has been improved, I hope it is clearer and more understandable. Description has been changed: ,,Multistep preparation of cellulose-filled chitosan films. i) the chemical structure of chitosan used to generate flexible and transparent films. ii) the structure of the two cellulose fillers used to reinforce chitosan, namely microcrystalline cellulose (MCC) with tactoids and isolated tiny crystals of nanocellulose (CNC). iii) the use of two different phpsphorylated reagents (POCl3 or N3P3Cl6) affords two kinds of phosphorylated cellulose (P-MCC / P-CNC and PN-MCC/PN-CNC). iv) mixing chitosan and cellulose derivative followed by the introduction of the resulting solution in a petridish and further evaporation of the solvent. v) digital photos of the resulting cellulose-filled-chitosan films illustrating their transparency”

  1. All charts must be aligned - one per row. Also, please use GraphPad or originlab software to create your charts

author response: The charts have been improved, I hope they are clearer and easier to read.
